# Corrosion-Induced Cracking Model of Concrete Considering a Transverse Constraint

**DOI:** 10.3390/ma17133217

**Published:** 2024-07-01

**Authors:** Xinrong Yan, Ye Tian, Dongming Yan, Litan Pan, Qiujing Zhou, Guoyi Zhang, Liang Pei, Xiang Lu, Bo Jiang, Weifeng Pan, Daquan Wang, Bin Chen, Yiran Li, Lin Luo

**Affiliations:** 1Huadian Electric Power Research Institute Co., Ltd., Hangzhou 310058, China; xinrong-yan@chder.com (X.Y.); litan-pan@chder.com (L.P.); daquan-wang@chder.com (D.W.); yiran-li@chder.com (Y.L.); 2Department of Civil Engineering and Architecture, Zhejiang University, Hangzhou 310058, China; dmyan@zju.edu.cn; 3China Institute of Water Resources and Hydropower Research, Beijing 100038, China; zhouqj@iwhr.com; 4College of Landscape Architecture, Zhejiang Agricultural and Forestry University, Hangzhou 310058, China; 5State Key Laboratory of Hydraulics and Mountain River Engineering, Sichuan University, Chengdu 610044, China; pl_scu@scu.edu.cn; 6College of Water Resources & Hydropower, Sichuan University, Chengdu 610065, China; scu-lx@scu.edu.cn; 7Powerchina Huadong Engineering Corporation, Hangzhou 310058, China; jiang_b@hdec.com; 8Guangzhou South Surveying and Mapping Technology Co., Ltd., Guangzhou 510640, China; pwf@southgnss.com; 9China Huadian Co. Ltd. Quzhou Wuxijiang Company, Quzhou 310030, China; windtalker2008@163.com; 10Huadian Yunnan Power Generation Co., Ltd., Kunming 650228, China; lin-luo@chd.com.cn

**Keywords:** transverse constraint, stirrup, three-layer hollow cylinder model, corrosion-induced cracking, critical corrosion rate

## Abstract

The performance of corrosion-induced cracking of reinforced concrete members under transverse constraints was studied. Based on the theory of elastic-plastic mechanics and the hypothesis of uniform corrosion of a steel bar, a three-layer hollow cylinder model was established to predict the critical corrosion of the steel bar at the time of the cracking of the concrete cover. Taking the constraint of stirrups on surrounding concrete into consideration, it can be used to predict the corrosion rate of members with stirrups at the time of the cracking of the concrete cover, which further expands the application range of the corrosion-induced cracking models of concrete. On this basis, the critical corrosion rate of concrete under different stirrup ratios at the time of cracking was measured. The calculated results of the model are in accordance with experimental data. For corner steel bars, when the stirrup spacing is less than 100 mm, the existence of stirrups can effectively delay the occurrence of rust expansion cracks and enhance the durability of the structure. On the basis of this study, the problem of corrosion expansion and cracking of the concrete cover caused by non-uniform corrosion of steel bars along longitudinal and radial directions needs to be further studied in the future.

## 1. Introduction

For a long time, reinforced concrete has been a meticulously designed material, as steel bars are well protected against corrosion by the alkaline environment provided by concrete. Therefore, in accordance with the Code for Design of Concrete Structures and other worldwide standards, reinforced concrete structures are supposed to provide good service performance for at least 50 years in China and other countries worldwide [1]. However, the material components of concrete react with CO_2_ and water in natural environments, which may reduce the alkalinity of the concrete [2]. Meanwhile, the chloride ions found in marine environments also diffuse into concrete and induce corrosion of steel bars. So the possibility of the corrosion of steel bars in concrete increases over time [3]. The volume of corrosion products generated by corrosion is 2–4 times larger than that of the original reinforcement [4]. The corrosion products continually grow after filling the gap between the steel bar and the concrete, creating pressure on the inner surface of the concrete cover [5,6,7,8,9], that is, corrosion-induced expansion force. With the increase in corrosion-induced expansion force, the concrete cover cracks from the inside to the outside [10]. The cracks induced by corrosion provide a convenient transport channel for the invasion medium from the external environment, resulting in further corrosion and deterioration of the steel reinforcement [11]. Currently, the corrosion of steel bars is one of the most important reasons for the deterioration of reinforced concrete structures [12,13,14,15].

Scholars have conducted a lot of research on the problem of concrete cracking caused by steel corrosion [16,17,18,19,20]. Andrade et al. [21] conducted an accelerated corrosion-induced cracking test and summarized various factors affecting the cracking of a concrete cover. Liu et al. [22] divided the process of cracking into the free expansion of corrosion products, the internal force generated on the inner surface of a concrete cover, and the cracking of the cover, which was widely accepted by scholars. Based on the premise that concrete is an orthotropic material, Pantazopoulou et al. [23] established the mathematical model of corrosion of steel bars at cracking time through theoretical deduction. Wang et al. [24,25] and Wang et al. [26] established elastic models to predict the cracking process of a concrete cover by assuming incompressible corrosion products. Coronelli et al. [27] regarded the concrete cover as a beam under uniform load and the stirrups as the elastic support of the beam and analyzed the corrosion-induced expansion force when cracks appeared. By comparing the previous research, it can be found that the existing theoretical models mostly adopt the single-layer cylinder model, ignoring the possibility of gradient changes in materials or cracking processes. However, among the factors that lead to the cracking caused by corrosion along the longitudinal direction, few scholars consider the influence of stirrups. Actually, the stirrups provide transverse constraint on the concrete and homogenize the corrosion-induced expansion force around the longitudinal bar [28], increasing the minimum value of the force that can cause cracks in a concrete cover. Moreover, the shorter the spacing between stirrups, the more obvious the constraint effect. Therefore, it is necessary to consider the transverse constraint of stirrups in the corrosion-induced cracking model of concrete.

In conclusion, most of the models in the existing research are single-layer cylinder models, and a few adopt multi-layer cylinder models. Few scholars have considered the effect of stirrups on the cracking of concrete covers caused by longitudinal reinforcement corrosion. In fact, stirrups provide circumferential constraints on concrete to inhibit concrete cracking, so it is necessary to consider the influence of stirrups in the rust expansion cracking model. To solve these problems, this paper mainly discusses the performance of corrosion-induced cracking of reinforced concrete members under the transverse constraint of stirrups. A three-layer hollow cylinder model is established, which is a theoretical model for the moment of concrete cover cracking. The model is verified by experimental data. By analyzing the relationship between the spacing of stirrups and the critical corrosion rate when a concrete cover cracks, the necessity of considering the constraint effect of stirrups is further demonstrated. After that, the critical stirrup spacing of the model is determined using finite element analysis software simulations.

## 2. Three-Layered Hollow Cylinder Model for Uniform Corrosion

### 2.1. Establishment of the Three-Layered Hollow Cylinder Model

Based on elastic mechanics, a corrosion cracking model was established to assess the critical corrosion ratio of the longitudinal steel bar that induced cracking, in which the influence of stirrups was taken into account. The model was demonstrated in Figure 1 in detail.

As shown in Figure 1a,b, the longitudinal steel bar can be classified into two types according to the position of the steel bar in the stirrup cage: the steel bar at the corner of the stirrup cage and the steel bar at the middle of the stirrup cage. Based on the symmetry, the constraint of stirrups and concrete around the steel bar at the corner of the stirrup cage can be divided into four parts: Parts I, II, III, and IV, as shown in Figure 1a. The steel bar in Parts II and III is simplified as a cylinder in semi-infinite space according to elastic mechanics, while the steel bar in Parts I and IV can be seen as a cylinder in finite space. The constraint of stirrups and concrete in Parts I and IV on the steel bar is obviously milder than that in Parts II and III. So the corrosion cracking of the steel bar at the corner of the stirrup cage usually appears in Parts I and IV. A similar analysis can be carried out on the longitudinal steel bar in the middle of the stirrup cage. In this situation, Parts II, III, and IV will apply a stronger contraint to the steel bar than in Part I. As a result, the steel bar in the middle of the stirrup cage in Part I is more susceptible to cracking. For simplicity, the easy-to-crack part of the concrete and stirrup around a longitudinal steel bar can be multiplied and folded to replace the other parts to model the critical corrosion cracking of reinforced concrete, which can be seen in Figure 1c. From the inside to the outside, the simplified model is composed of a steel bar, stirrups, and concrete. It should be noticed that there is a small amount of concrete between the longitudinal steel bar and the stirrup. Therefore, the regional replacement can be further simplified to establish a three-layered hollow cylinder model to characterize the critical cracking induced by the corrosion of the longitudinal steel bar, as shown in Figure 1d. At the center of the three-layered cylinder model, the original longitudinal steel bar is represented as a circular void. The stirrup is equivalent to a thin layer evenly distributed longitudinally around the void. The volume of the stirrup layer is the same as that of the spaced stirrups, and the center line of the cylinder coincides with the centroid of the stirrup. The inner concrete layer is defined as the concrete between the stirrup layer and the longitudinal steel bar, and the outer concrete layer is equivalent to the concrete cover. Based on the three-layered model, the uniform corrosion-induced expansion state is simplified to the plane strain state. The radial displacement caused by the corrosion-induced expansion force can still be solved by elastic mechanics. In this research, the corrosion of stirrups is ignored for simplicity, and the corrosion of the longitudinal steel bar is assumed to be uniform.

### 2.2. Theoretical Derivation

Based on elastic mechanics, the analytical solution of the corrosion-induced expansion stress can be obtained by applying the corrosion expansion force to the surface of the circular void. The theoretical derivation of the model is divided into the following three steps: Firstly, the stress and strain of the cylinder under uniform loading on the inner surface of the circular void are calculated. Secondly, the stress and strain of the cylinder corresponding to the initiation and transfixion of corrosion cracking are solved analytically. Finally, by analyzing the stress of corrosion, the corrosion ratio during cracking of the longitudinal bar is given.

#### 2.2.1. Non-Cracking Stage

At the non-cracking stage, all three layers are crack-free. The corrosion-induced expansion stress is applied to the inner surface of the model. The stress of the concrete cover in the elastic stage is shown in Figure 2. *R*_0_ is the inner radius of the first layer, that is, the radius of the corroded longitudinal steel bar, including the corrosion products on it. *R*_1_ is the outer radius of the first layer, that is, the distance from the centroid of the longitudinal steel bar to the outer surface of the inner concrete layer. *R*_2_ is the outer radius of the second layer, that is, the distance from the center to the outer surface of the stirrup layer. *R*_3_ is the outer radius of the third layer, that is, the distance from the center to the outer surface of the concrete cover.

Here, *R*_1_ and *R*_2_ can be given as:(1)$$\begin{array}{l} R_{1}=R_{0}+R_{g}-\pi R_{g}^{2}/(2s)\\ R_{2}=R_{0}+R_{g}+\pi R_{g}^{2}/(2s) \end{array}$$
where *R_g_* is the radius of the stirrup and *s* is the spacing between the stirrups.

Polar coordinates are used to solve the equations. When the corrosion-induced expansion force is small, the concrete is in the elastic state, and the internal force is solved by the state-space method. The geometric equation is shown below:(2)$$\begin{array}{l} \varepsilon_{rr,i}=du_{r,i}/dr\\ \varepsilon_{\theta \theta ,i}=u_{r,i}/r \end{array}$$
where *ε_rr_*_,__*i*_ and *ε_θθ_*_,__*i*_ are the radial strain and the circumferential strain of layer *i*, respectively, and *u_r_*_,__*i*_ is the radial displacement at the radius of *r* in layer *i*.

The constitutive equations are defined as:(3)$$\begin{array}{l} \sigma_{rr,i}=c_{11,i}(du_{r,i}/dr)+c_{12,i}(u_{r,i}/r)\\ \sigma_{\theta \theta ,i}=c_{12,i}(du_{r,i}/dr)+c_{22,i}(u_{r,i}/r) \end{array}$$
where *σ_rr_*_,__*i*_ and *σ_θθ_*_,__*i*_ are the radial stress and the circumferential stress of layer *i*, respectively, and *c_lm_*_,__*i*_ is the elastic constant of layer *i*. To simplify the calculation, the Poisson’s ratio is taken to be 0. Then *c*_11,__*i*_ = *E_i_*/2, *c*_12,__*i*_ = 0, and *c*_22,__*i*_ = *E_i_*, where *E_i_* is the elastic modulus of layer *i*.

The differential equation of equilibrium is shown below:(4)$$\frac{d\sigma_{rr,i}}{dr}+\frac{\sigma_{rr,i}-\sigma_{\theta \theta ,i}}{r}=0$$

The nondimensional numbers *c*_11*D*,*i*_ = *E_i_*/*E*_1_, *c*_12*D*,*i*_ = 0, *c*_22*D*,*i*_ = 2*E_i_*/*E*_1_, *u_i_* = *u_r_*_,__*i*_/*R*_3_, *σ_r_*_,__*i*_ = *σ_rr_*_,__*i*_/*σ*_11,1_, *σ_θ_*_,__*i*_ = *σ_θθ_*_,__*i*_/*σ*_11,1_, *ξ* = *r*/*R*_3_, and *ξ_i_* = *R_i_*/*R*_3_ are introduced into the calculation. The simplified constitutive equations are deduced as:(5)$$\begin{array}{l} \Sigma_{r,i}=c_{11D,i}\nabla u_{i}+c_{12D,i}u_{i}\\ \Sigma_{\theta ,i}=c_{12D,i}\nabla u_{i}+c_{22D,i}u_{i} \end{array}$$

The differential equation of equilibrium is shown in Equation (6).
(6)$$\nabla \sum_{r,i}-\sum_{\theta ,i}=0$$
where Σ*_r_*_,__*i*_ = ξ*σ_r_*_,__*i*_, Σ*_θ_*_,__*i*_ = ξσ*_θ_*_,__*i*_, and ∇ = ξ∂/∂ξ.

The boundary conditions are given as:(7)$$\begin{array}{l} \sum_{r,i}(\xi_{0})=\xi_{0}q/c_{11,1}=\xi_{0}p\\ \sum_{r}^{3}(\xi_{3})=\xi_{3}q_{3}/c_{11,1}=0 \end{array}$$
where *q* (N/mm^2^) is the corrosion-induced expansion force of the steel bar and *q*_3_ is the normal stress on the outer surface of the concrete cover, whose value is 0.

For the convenience of calculation, the constitutive equations and the differential equations of equilibrium are expressed in matrix form, as shown in Equation (8).
(8)$$\begin{array}{l} \nabla \left[X_{i}(\xi )\right]=\left[N_{i}\right]\left[X_{i}(\xi )\right],\\ \sum_{\theta ,i}=a_{3,i}u_{i}(\xi )+a_{4,i}\sum_{r,i}(\xi ) \end{array}$$
where $\left[X_{i}(\xi )\right]=\left[\begin{array}{c} u_{i}(\xi )\\ \sum_{r,i} \end{array}\right]$, $\left[N_{i}\right]=\left[\begin{array}{cc} a_{1,i} & a_{2,i}\\ a_{3,i} & a_{4,i} \end{array}\right]$, $a_{1,i}=-c_{12D,i}/c_{11D,i}$, $a_{2,i}=1/c_{11D,i}$, $a_{3,i}=a_{1,i}c_{12D,i}+c_{22D,i}$, and $a_{4,i}=a_{2,i}c_{12D,i}$.

The solution of Equation (8) can be shown as follows:(9)$$\left[X_{i}(\xi )\right]=\left[T_{i}(\xi )\right]\left[X_{i}(\xi_{i-1})\right]$$

Each term in the solution is demonstrated as follows:(10)$$\begin{array}{cc} \left[T_{i}(\xi )\right]= & F_{0,i}(\xi )I+F_{1,i}(\xi )\left[N_{i}\right],\\ F_{0,i}(\xi )= & \left[\lambda_{2,i}{\left(\xi /\xi_{i-1}\right)}^{\lambda_{1,i}}-\lambda_{1,i}{\left(\xi /\xi_{i-1}\right)}^{\lambda_{2,i}}\right]/(\lambda_{2,i}-\lambda_{1,i}),\\ F_{1,i}(\xi )= & \left[{\left(\xi /\xi_{i-1}\right)}^{\lambda_{2,i}}-{\left(\xi /\xi_{i-1}\right)}^{\lambda_{1,i}}\right]/(\lambda_{2,i}-\lambda_{1,i}) \end{array}$$
where *I* is the identity matrix and *λ*_1,__*i*_ and *λ*_2,__*i*_ are the characteristic roots of [*N_i_*].

The continuity condition $\left[X_{i+1}(\xi_{i})\right]=\left[X_{i}(\xi_{i})\right]$ is reused. It shows that the stress and displacement components of the outer surface of layer *i* are equal to the stress and displacement components of the inner surface of layer *i* + 1 of the cylinder.

The solution is $\left[X_{i}(\xi_{i})\right]=\left[H_{i}\right]\left[X_{1}(\xi_{0})\right]$, where $\left[H_{i}\right]=\left[T_{i}\right]\left[H_{i-1}\right]$ and [*H*_0_] is the identity matrix.

The boundary conditions expressed in Equation (7) are used to solve the problem, and the results are shown as follows:(11)$$u_{r,1}(R_{0})=-\xi_{0}(H_{22,3}/H_{21,3})(q/E_{1})$$
(12)$$\begin{array}{l} u_{r,i}(r)=\xi_{0}(H_{12,i-1}-\frac{H_{22,3}}{H_{21,3}}H_{11,i-1})T_{11,i}(\frac{r}{R_{3}})(q/E_{1})\\ -\xi_{0}(H_{22,i-1}-\frac{H_{22,3}}{H_{21,3}}H_{21,i-1})T_{12,i}(\frac{r}{R_{3}})(q/E_{1}) \end{array}$$

#### 2.2.2. Cracking Stage

With the increase in corrosion-induced expansion force, the corrosion cracking will initiate from the surface of the longitudinal steel bar and grow outward. In the cracking stage, the state and stress distribution in the concrete and stirrup are demonstrated in Figure 3. *R* is the distance from the crack tip to the center of the three-layered cylinder. It is clear that elastic mechanics is not appropriate to describe the nonlinear behavior of cracking concrete. In this study, the corrosion cracks are dispersed uniformly in the cracked concrete, and the analytical calculation of Zhou’s method [29] is adopted. It is assumed that the circumferential stress of the concrete is distributed in a triangle along the thickness of the concrete cover, as shown in Figure 3b.

The radial stress of the concrete on the inner surface of the first layer, *A*, can be transferred from *q* as follows:(13)$$A=\xi_{0}(q/E_{1})$$

Equation (2) is substituted into Equation (11) so the circumferential strain, *ε_θθ_*_,__*i*_, can be given as:(14)$$\begin{array}{l} \varepsilon_{\theta \theta ,i}(r)=\frac{A}{r}(H_{12,i-1}-\frac{H_{22,3}}{H_{21,3}}H_{11,i-1})T_{11,i}(\frac{r}{R_{3}})\\ -\frac{A}{r}(H_{22,i-1}-\frac{H_{22,3}}{H_{21,3}}H_{21,i-1})T_{12,i}(\frac{r}{R_{3}}) \end{array}$$

As demonstrated in Figure 2, the cracking of the inner concrete layer is restrained by the stirrups, and the cracking of this thin layer shows no remarkable influence on the bearing capacity of the concrete beam. In this research, the corrosion cracking is crucial only when the cracks propagate to the outer concrete layer. At the tip of the corrosion cracks, the cracking strain of the concrete can be determined by the tensile strength and elastic modulus of the concrete. So at the tip of the cracks, the circumferential strain can be calculated as:(15)$$\varepsilon_{\theta \theta ,3}(R)=\frac{f_{t}}{E_{c}}$$
where *f_t_* is the tensile strength of the concrete and *E_c_* is the elastic modulus of the concrete, which is the assignment of *E*_1_ and *E*_3_.

Equations (14) and (15) are combined, so parameter *A* can be solved.
(16)$$A=\frac{f_{t}R}{E_{c}\left[\left(H_{12,i-1}-\frac{H_{22,3}}{H_{21,3}}H_{11,i-1})T_{11,i}(\frac{R}{R_{3}})-(H_{22,i-1}-\frac{H_{22,3}}{H_{21,3}}H_{21,i-1})T_{12,i}(\frac{R}{R_{3}}\right)\right]}$$

Based on the three-layered hollow cylinder model, the longitudinal length of the reinforced concrete is taken as 1. Then, the tension force *F_c_*_c_ provided by the cracked part of the concrete is obtained by analyzing the quarter of the cylinder, as shown in Figure 3b.
(17)$$F_{cc}=\frac{1}{2}f_{t}[R-R_{0}-\frac{R_{1}+R_{2}-2R_{0}}{R-R_{0}}(R_{2}-R_{1})]$$

The tensile force *F_c_*_u_ provided by the uncracked part of the concrete is derived as follows:(18)$$F_{cu}=E_{c}\int_{R}^{R_{3}}\varepsilon_{\theta \theta ,3}(r)dr$$

The tensile force *F_st_* provided by the equivalent stirrup is shown as follows:(19)$$F_{st}=E_{st}\int_{R_{1}}^{R_{2}}\varepsilon_{\theta \theta ,2}(r)dr$$
where *E_st_* is the elastic modulus of the stirrup, which is the assignment of *E*_2_.

Here, the cracking force *F* is defined as the sum of *F_c_*_c_, *F_c_*_u_, and *F_st_*. The radius *R* from the crack tip to the cylinder center at the moment of cracking is calculated by using *∂F*/*∂R* = 0. The radial displacement *u_r_*_,1_(*R*_0_) on the inner face of the inner concrete layer can be obtained by Equation (11).

#### 2.2.3. Internal Force Analysis of Corrosion Products

In this research, the corrosion of the longitudinal steel bar is assumed to be uniform. The corrosion products will accumulate on the surface of the corroded steel bar to form a thin layer. The volume growth of the corrosion products will generate an expansion force on the concrete around the steel bar. At the same time, the acting force will perform compression on the corrosion products, which will cause volume shrinkage. The volume change in the corroded steel bar is demonstrated in Figure 4. *R_s_*_0_ is the original radius of the steel bar; *R_s_*_1_ is the radius of the residual uncorroded steel bar; *R_s_*_2_ is the radius of the steel bar after the free expansion of the corrosion products; and *R_s_* is the radius of the steel bar constrained by the surrounding concrete. Zhao et al. [30,31] and Michel et al. [32] proved that the corrosion products will not fill the cracks before the surface cracking of the concrete cover.

Lundgren [33] proposed the constitutive relationship of the corrosion products as shown below:(20)$$q_{L}=K_{cor}\varepsilon_{cor}^{p}$$
where *ε_cor_* is the radial compressive strain of the corrosion products; *q_L_* is the corresponding radial compressive stress; and *K_cor_* and *p* are the material parameters of the corrosion products.

In previous studies, *K_cor_* and *p* were proposed as 7 and 7 × 10^7^ Pa [33,34], respectively. The radial strain of the corrosion products can be obtained by substituting the value of *q_L_* into Equation (18).

#### 2.2.4. Corrosion Rate of the Steel Bar at the Moment of Cracking

There has been some research on the cracking corrosion ratio of a steel bar when the concrete cover cracks [22,35,36]. Among them, Liu et al. [22] proposed a three-stage model to describe the expansion of corrosion products in uncracked and cracked reinforced concrete, which was widely accepted by scholars. Based on their research, the corrosion products will not produce a corrosion-induced expansion force before the interfacial gap between the steel bar and the concrete is completely filled. Also, the corrosion products will not flow into the corrosion-induced cracks until the cracks reach the surface of the concrete cover. As a result, the interfacial gap between the steel bar and the concrete is simplified as a thin void layer in the model analysis, and the corresponding thickness, *d*, is set at 0.012 mm [22]. By taking into account the influence of the void layer on the volume compression of the corrosion products, the corrosion depth of the steel bar and the thickness of the corrosion products, *x_cor_* and *d_cor_*, are given as:(21)$$x_{cor}=\frac{\rho R_{s0}}{2}$$
(22)$$d_{cor}=\frac{\rho R_{s0}}{2}+d+u_{r,1}(R_{0})$$

Then, the radial compressive strain of the corrosion products, *ε_cor_,* is derived based on the geometric characteristics of the corroded steel bar as follows:(23)$$\frac{nR_{s0}\rho /2-\left[u_{r,1}\left(R_{0}\right)+d+R_{s0}\rho /2\right]}{nR_{s0}\rho /2}=\varepsilon_{cor}$$
where *n* is the volume expansion ratio of the corrosion products and *ρ* is the corrosion ratio of the longitudinal steel bar.

### 2.3. The Influence of Spacing between Stirrups on the Critical Corrosion Rate

According to the model established above, a numerical example is constructed. The parameters are shown in Table 1.

The concrete cover is 15 mm thick, the diameter of the longitudinal bar is 16 mm, the diameter of the stirrup is 6 mm, and the spacing between the stirrups increased from 25 mm to 400 mm. The relationship between the critical corrosion-induced expansion force *q_L_* and the spacing between stirrups is shown in Figure 5. The relationship between the critical corrosion rate *ρ_L_* and the spacing between stirrups is shown in Figure 6.

It can be seen from the figures that both *q_L_* and *ρ_L_* decrease as the spacing between the stirrups increases. When the spacing between the stirrups is more than 100 mm, both *q_L_* and *ρ_L_* remain almost unchanged. When the spacing between the stirrups is less than 100 mm, both *q_L_* and *ρ_L_* increase sharply as the spacing between the stirrups decreases.

It follows that when the spacing between stirrups is greater than 100 mm, the spacing between stirrups is no longer the main factor controlling *q_L_* and *ρ_L_*. When the spacing between stirrups is less than 100 m, the shorter the spacing between stirrups, the more obvious the constraint effect. As a result, the values of *q_L_* and *ρ_L_* become higher and higher.

In conclusion, when the spacing between stirrups is less than 100 mm, the transverse constraint of the stirrups can effectively delay the occurrence of cracks in the concrete, thus improving the durability of the overall reinforced concrete structure.

## 3. Test Process

In order to verify the accuracy of the model and study the effect of the stirrup ratio on the critical corrosion of a steel bar, an experiment was designed. In this experiment, the spacing between stirrups and the type of longitudinal bar were considered.

### 3.1. Specimens Design

A total of 24 specimens were designed for this experiment, as shown in Table 2. Figure 7 depicts the geometric dimensions of the specimens and how the rebars are placed. Some of the longitudinal bars are ribbed steel bars of HRB400 with a diameter of 16 mm, and the others are plain round bars of type HPB300 with a diameter of 16 mm. The stirrups are plain round bars of type HPB300 with a diameter of 6 mm. The thickness of the concrete cover is 15 mm.

The concrete mixture rate is shown in Table 3. The compressive strength of the concrete after curing for 28 days is shown in Table 4. The mechanical properties of the rebars are shown in Table 5.

The results show that the elastic modulus of the HRB400 ribbed steel bar is higher than the empirical value. The main reason for this phenomenon is that the ribbed steel bar is irregular in shape and its cross-sectional area cannot be accurately measured, so it is difficult to accurately calculate the normal stress of the section of steel bar and elastic modulus.

### 3.2. Accelerated Corrosion Method

In order to simulate the corrosion of the steel bar, a method of accelerating galvanic corrosion was used. The galvanic accelerated corrosion method is shown in Figure 8. The concrete block is covered with a sponge, and the sponge is secured with stainless-steel mesh. A layer of insulating plastic cloth covers the outside of the stainless-steel mesh to ensure that the concrete remains wet during the test. The positive pole of the DC power supply is connected to the exposed steel bar, and the negative pole is connected to the stainless-steel mesh. The reason for this connection is that electrons flow from the negative to the positive side of DC power. As the anode, the steel bar loses electrons, so the electrons enter the stainless-steel mesh and flow out from the exposed section of the steel.

The concrete block needs to be immersed in a 5% NaCl solution for three days before electrification to make the concrete fully wet. At the same time, in order to prevent water runoff and electricity leakage, the surface of the stainless-steel mesh needs to be sealed with a waterproof and insulating plastic cloth, and a 5% NaCl solution needs to be added every day to ensure its moisture. The curing of the specimens and the test site are shown in Figure 9.

In the process of galvanizing corrosion, the concrete crack observation instrument is used to observe the surface of the specimens twice a day, in the morning and evening. Once cracks appear, the DC power is stopped immediately, and the cover of the concrete is shattered to remove the steel bar. Finally, the rust on the surface of the steel bar is removed, and the mass loss is measured to calculate the corrosion rate.

In this experiment, chemical methods and mechanical methods are combined to remove the rust. First, the ZJ-828 rust remover is used. It can dissolve most of the corrosion products and loosen the remaining undissolved corrosion products. Then, the remaining corrosion products are removed mechanically. As shown in Figure 10, the descaling effect is satisfactory.

### 3.3. Analysis of Test Results

The critical corrosion rate obtained by the model is compared with the actual test results (see Table 6 and Table 7 for details). Since the diameter, stirrup ratio, mechanical properties, and other indicators of the ribbed steel bars and plain round bars are exactly the same, their calculated critical corrosion rates are the same, all around 0.641~0.832%. When the concrete cover cracks, the measured critical corrosion rate of the plain round bar is around 0.628~0.884%. For the ribbed steel bars, the measured critical corrosion rate is around 0.651~0.810%. It was found that the actual measured corrosion rate is discrete. The main reason is that the number of specimens is too small. However, since the deviation is basically within 20%, the calculations are still convincing. In this test, the longitudinal bar is always free from load during the process of corrosion. By comparing the actual measured data in two tables, it can be seen that when the longitudinal bar is not loaded, the critical corrosion rate of the concrete with the plain round bar is basically the same as that of the concrete with the ribbed steel bar.

By comparing the data in Table 6 and Table 7, it can be found that when there is no stirrup in the concrete, the model can almost fully explain the actual state. The same situation happens when the spacing between the stirrups is about 50 mm or less. This shows that the assumptions of the model and its theoretical derivation are correct. When the spacing between the stirrups was between about 50 mm and 100 mm, the model deviated slightly from the actual state. This is because the model assumes that the stirrup is continuous longitudinally. If the spacing between stirrups is too large, the assumption will not hold. This critical value of the spacing is discussed in detail in the finite element analysis section.

In order to consider the relationship between the critical corrosion depth of the steel bar and the thickness of the concrete cover, the diameter of the steel bar, and the tensile strength of the concrete when the concrete cover cracks, the calculation results of the model are compared with the experimental results obtained in some other scholars’ research [11,21,37]. Since the chemical composition of the corrosion products of steel bars is related to the environment around the steel bars, when calculating the corrosion amount of the steel bars at the time of the cracking of the concrete cover, the value of *n* is generally 2.5 or 3. The calculated results are shown in Table 8. It can be seen that the calculated results are greatly affected by the coefficient of expansion *n*. Even so, they are basically consistent with the results obtained by the experiments of various scholars. From this conclusion, it can be seen that the model proposed above can predict the critical corrosion of a steel bar at the time of the cracking of the concrete cover.

## 4. Analysis with the ABAQUS Finite Element Software

In the previous section, the critical value of spacing between stirrups was discussed. In practice, the constraint of each stirrup on the surrounding concrete is within certain limits and has no effect on the concrete outside the range. Therefore, only a single-layer cylinder model can be used for the concrete outside the constraint range. In order to make the application of the theoretical model meet the constraint conditions, it is necessary to determine the critical spacing between stirrups. The critical spacing between stirrups is in terms of *S_cr_*. When the spacing between the stirrups is less than *S_cr_*, all the longitudinal concrete will be constrained by the stirrups, and the three-layer hollow cylinder model is consistent with the actual state. Conversely, when the spacing between the stirrups is larger than *S_cr_*, the single-layer cylinder model is used for the concrete outside the constraint range, while the three-layer hollow cylinder model can still be used for the concrete within the constraint range. In this section, the finite element software ABAQUS will be used for simulation analysis, and the critical spacing between stirrups will be briefly discussed.

### 4.1. The Establishment of the ABAQUS Model

The length and width of the cross-section of the model are 100 mm. A 16 mm diameter HRB400 longitudinal bar is placed near the corner of the model. The thickness of the concrete cover is 15 mm. The spacing between stirrups is divided into four conditions: 100 mm, 70 mm, 50 mm, and 0 (without stirrup). The number of steel bars is shown in Table 9. The constitutive relation of the concrete is based on the plastic damage model of C55 concrete given in *Code for design of concrete structures GB50010-2010* [38]. A damage field is used to describe the track of the internal cracks in the concrete. The model adopts the stirrup of HPB300, whose elastic modulus is 2.1 × 10^5^ N/mm^2^, Poisson’s ratio *μ* is 0.3, and yield strength is 300 N/mm^2^. The type of finite element in this model is C3D8R. C3D8R is an eight-node linear hexahedral element, which is the least time-consuming hexahedral element in ABAQUS, so it is the most widely used in the practical application of a volume element [39]. The model contains 26500 elements, in which the element density is increased at the steel–concrete interface to improve the calculation accuracy. The size and mesh generation of the model are shown in Figure 11, and the position relationship between the longitudinal bar and stirrups is shown in Figure 12.

### 4.2. Critical Spacing between Stirrups

As shown in Figure 13, the concrete outside the constraint range *S_cr_* can be considered to have no stirrup internally, and its corrosion-induced expansion characteristics are the same as those of the member without a stirrup.

Both the theoretical model and the finite element software ABAQUS can calculate the ratios of the corrosion-induced expansion force of the specimens with stirrups and those without stirrups at the cracking time. Table 10 shows the ratio relationship between different spacings and different sectional areas of the stirrups.

Comparing the two results, it is found that the calculation results of ABAQUS are generally smaller than those of the theoretical model. The main reason is that the theoretical model assumes that the corrosion-induced expansion force is uniformly distributed. However, in ABAQUS, although the longitudinal bar is uniformly expanded, the uneven stress distribution in some parts will be caused by the different thicknesses of the surrounding concrete cover. These parts are prone to stress concentration. Therefore, the calculation results of the finite element software ABAQUS are generally smaller than those of the theoretical model.

By comparing the data in the table, it is found that when the spacing between the stirrups and the sectional area are both large, the calculated result of ABAQUS and the theoretical value deviate greatly. That is, the force calculated by ABAQUS is significantly less than the calculated value of the theoretical model. This indicates that the stirrups do not completely constrain the concrete, leading to premature cracking of the unconstrained parts of the concrete. As the spacing between the stirrups decreases, the deviation of the calculated result between ABAQUS and the theoretical model gradually decreases. This indicates that when the spacing between stirrups is less than a certain value, the stirrups can effectively constrain all the concrete around the longitudinal bars, and it is reasonable to equate the stirrups as cylinders of equal volume.

When the spacing between the stirrups is less than 100 mm, the deviation can be basically controlled within 30%. Therefore, it is considered that when the spacing between stirrups is less than 100 mm, the theoretical model meets the application conditions, and the three-layer hollow cylinder model is correct. When the spacing between the stirrups is greater than 100 mm, the stirrups have no constraint on the concrete 50 mm away from the stirrups. The concrete in those parts is modeled as a single-layer cylinder. Concrete within 50 mm of the stirrup is still treated as the three-layer hollow cylinder model. Therefore, in this model, the critical spacing between stirrups is 100 mm.

Since the volume expansion rate of the corrosion product has a great influence on the accuracy of this model, the volume expansion rate should be reasonably determined according to the chemical composition of the corrosion product in practical applications. The calculation model proposed in this paper assumes that the corrosion of steel bars is uniform and the corrosion-induced expansion force is uniformly distributed. The corrosion expansion cracking of the concrete cover caused by non-uniform corrosion of steel bars along the longitudinal and circumferential directions needs further study.

## 5. Conclusions

Based on the theory of elastic mechanics, the critical corrosion depth of a longitudinal bar in a reinforced concrete member with stirrups is analyzed, and a three-layer hollow cylinder model is established. The calculated results of the theoretical model are basically consistent with the experimental data, which shows the correctness of the model. The following conclusions were drawn:

(1) The rust expansion of steel bars causes radial displacement of the inner wall of the concrete cover, resulting in the surface of the concrete cover reaching its ultimate tensile strain and causing cracks. The cracked part of the concrete cover causes the uncracked concrete cover to crack. When the longitudinal bar is not loaded, the critical corrosion rate of the concrete with a plain round bar is basically the same as that of the concrete with a ribbed steel bar.

(2) For the steel bar in the concrete corner, when the spacing between stirrups is more than 100 mm, whether stirrups are added or not has little effect on the resistance of the reinforced concrete members to corrosion-induced cracking. When the spacing between stirrups is less than 100 mm, the transverse constraint of the stirrups can effectively delay the occurrence of cracks in the concrete, thus improving the durability of the reinforced concrete structure.

(3) Since the volume expansion rate of the corrosion product has a great influence on the accuracy of this model, the volume expansion rate should be reasonably determined according to the chemical composition of the corrosion product in practical applications.

(4) The calculation model proposed in this paper assumes that the corrosion of steel bars is uniform and the corrosion-induced expansion force is uniformly distributed. The corrosion expansion cracking of the concrete cover caused by non-uniform corrosion of steel bars along the longitudinal and circumferential directions needs further study.

## Figures and Tables

**Figure 1 materials-17-03217-f001:**
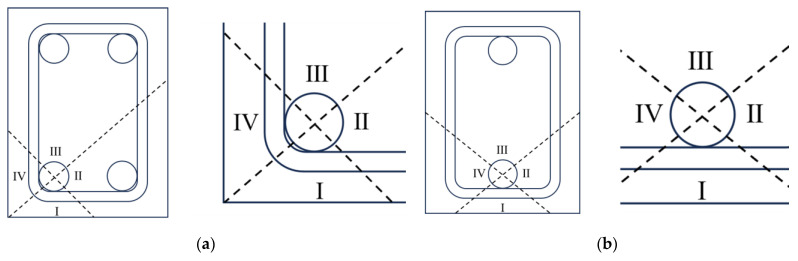
The simplified model demonstrating the critical cracking induced by the corrosion of the steel bar. (**a**) Division of concrete around the steel bar at the corner of the stirrup cage; (**b**) Division of concrete around the steel bar in the middle of the stirrup cage; (**c**) The result of regional replacement; (**d**) Three-layered hollow cylinder model.

**Figure 2 materials-17-03217-f002:**
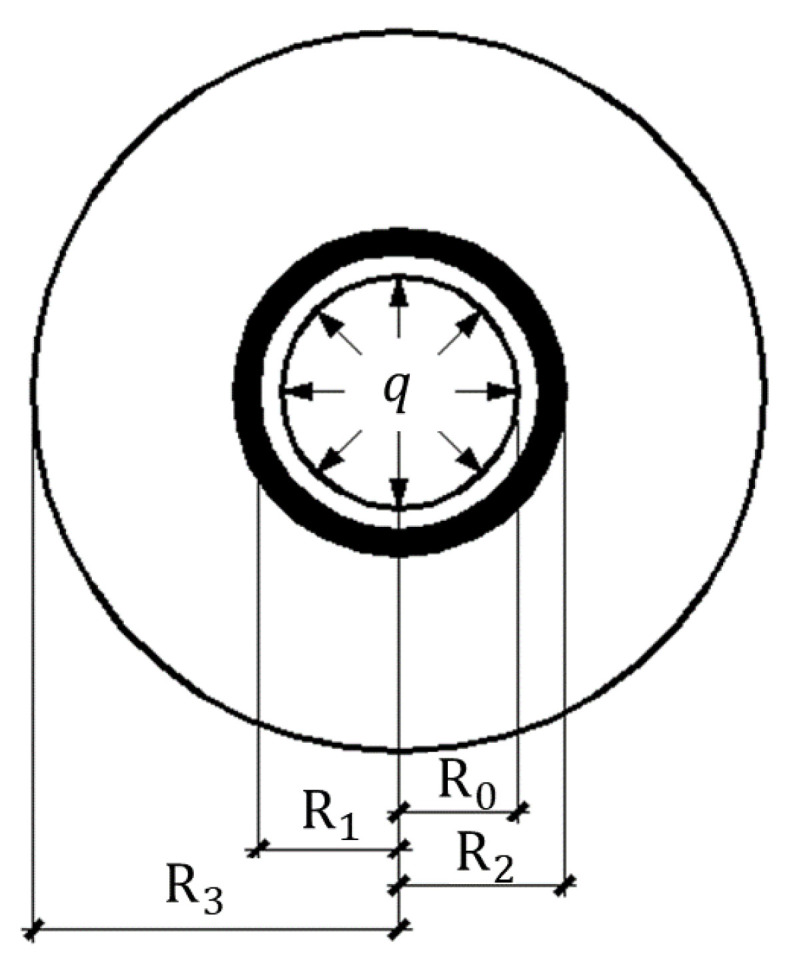
Stress state of the reinforced concrete in the non-cracking stage.

**Figure 3 materials-17-03217-f003:**
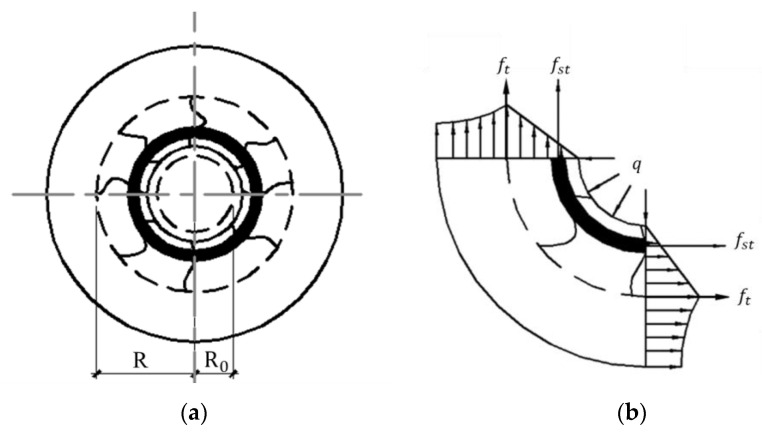
Stress analysis of the concrete cover during the cracking stage. (**a**) Concrete at cracking time, (**b**) Distribution of the circumferential stress of concrete.

**Figure 4 materials-17-03217-f004:**
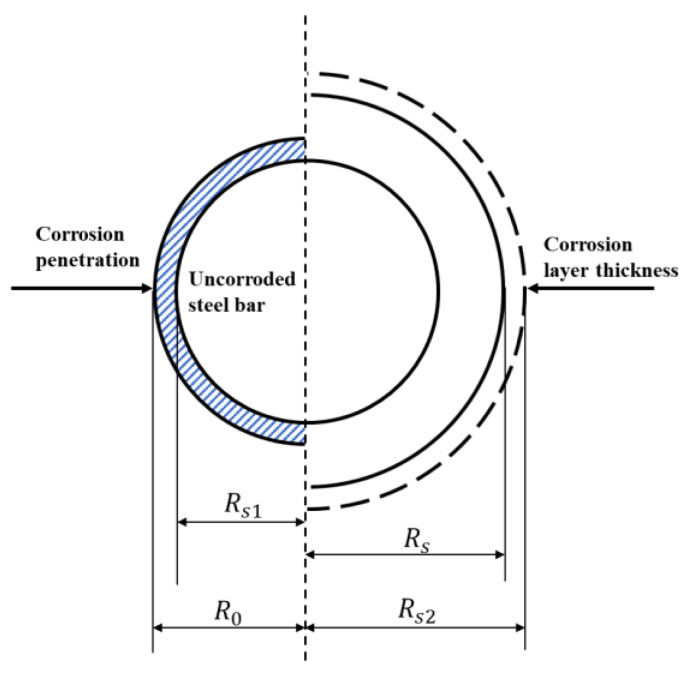
Expansion process of the corrosion products.

**Figure 5 materials-17-03217-f005:**
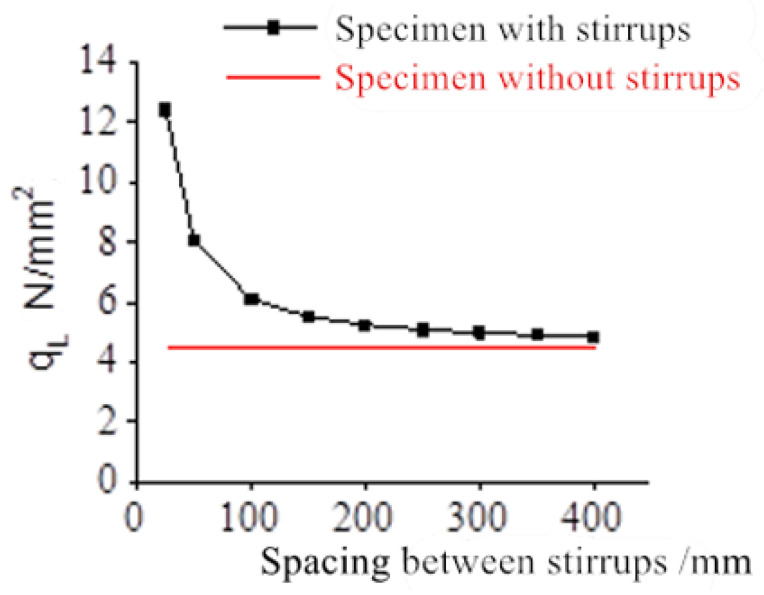
The relationship between the critical corrosion-induced expansion force *q_L_* and the spacing between stirrups.

**Figure 6 materials-17-03217-f006:**
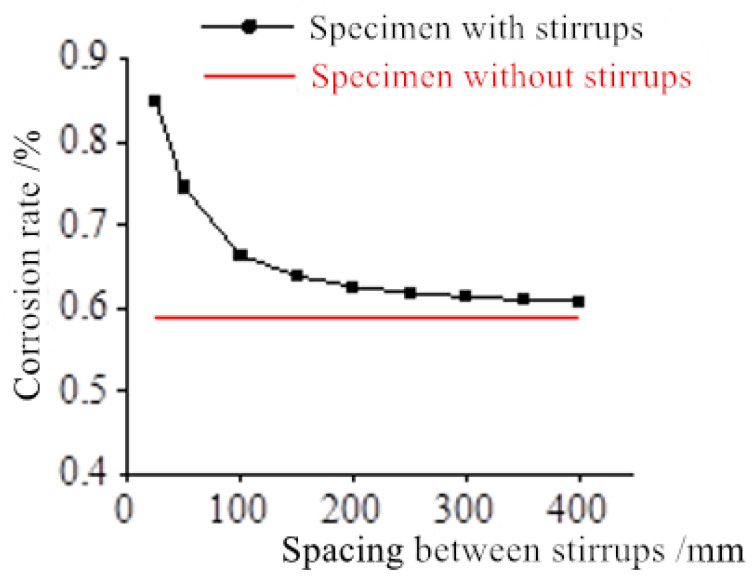
The relationship between the critical corrosion rate *ρL* and the spacing between stirrups.

**Figure 7 materials-17-03217-f007:**
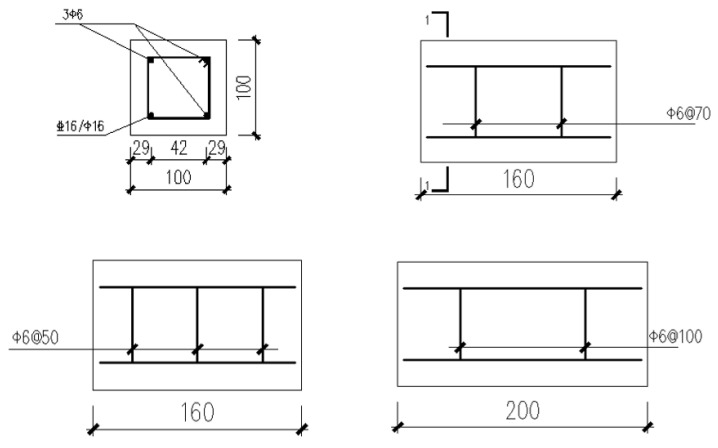
The geometric dimensions of the specimens and the reinforcement.

**Figure 8 materials-17-03217-f008:**
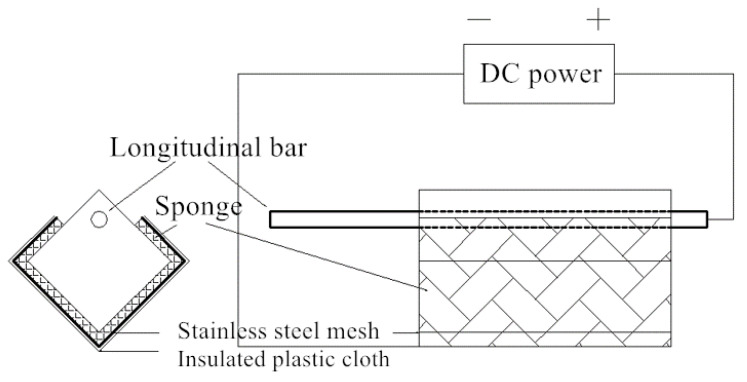
Accelerated galvanic corrosion test device.

**Figure 9 materials-17-03217-f009:**
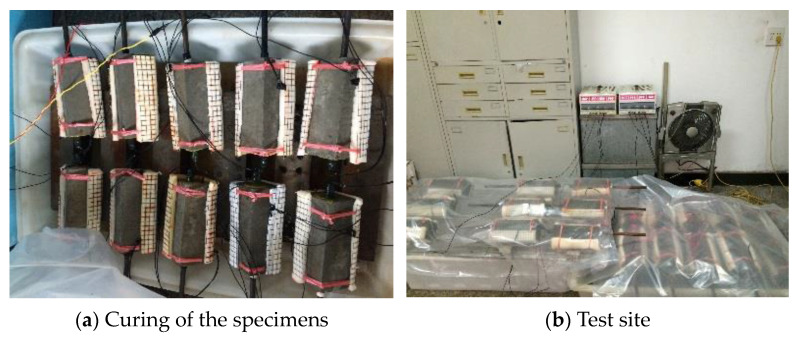
Accelerated galvanic corrosion test site.

**Figure 10 materials-17-03217-f010:**
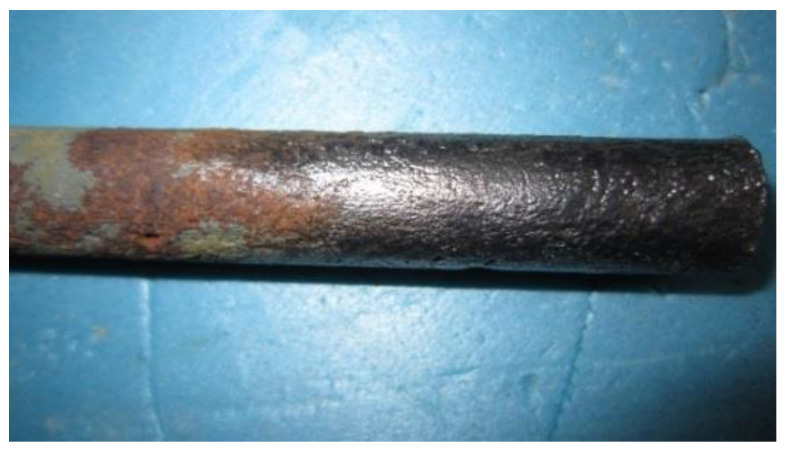
The effect before and after derusting.

**Figure 11 materials-17-03217-f011:**
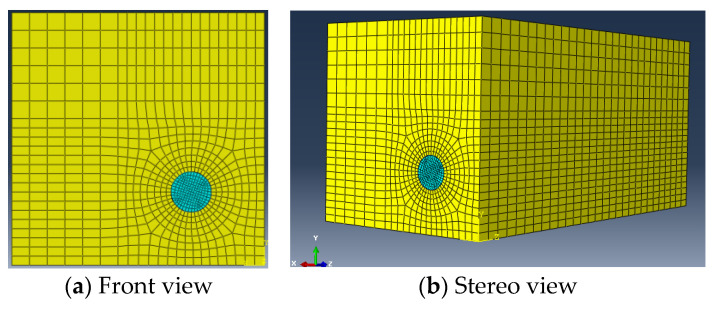
The size and mesh generation of the model.

**Figure 12 materials-17-03217-f012:**
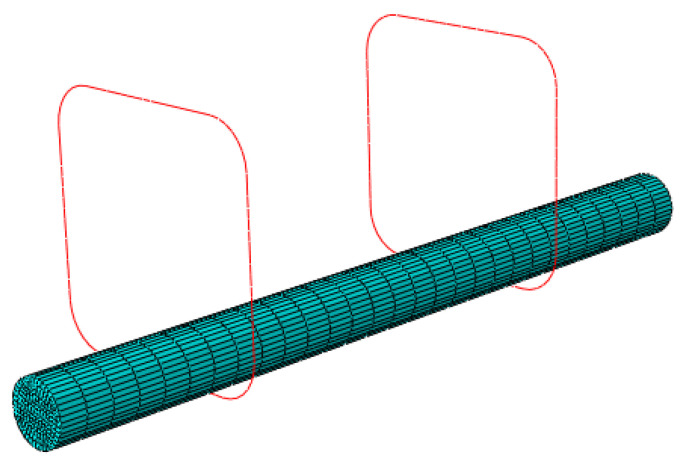
The position relationship between the longitudinal bar and stirrups.

**Figure 13 materials-17-03217-f013:**
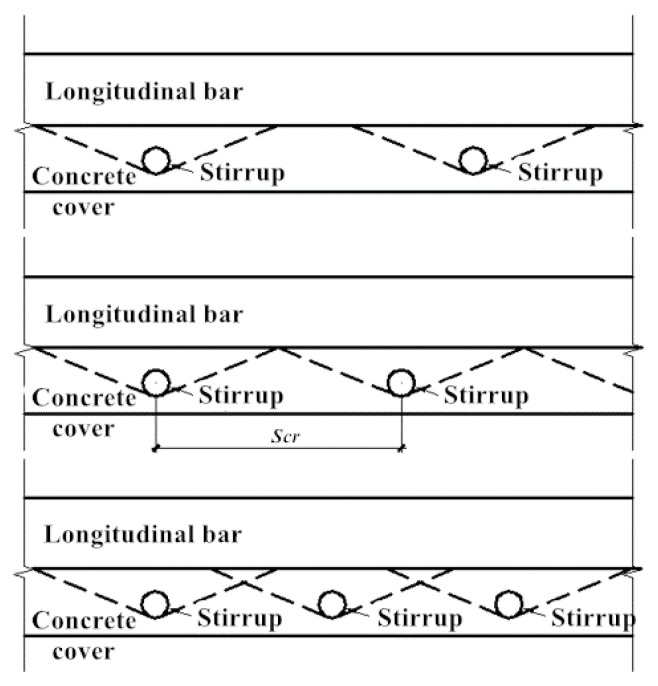
The constraint range of the stirrups on the concrete.

**Table 1 materials-17-03217-t001:** Material parameters.

Material	Type	Characteristic Value of Strength *f_ct_*, *f_st_*/N/mm^2^	Elasticity Modulus *E_c_*, *E_st_*/N/mm^2^	Expansion Rate of Corrosion Products *n*	Spacing between Stirrups *s*/mm
Concrete	C55	35.5	*E_c_* = 3.55 × 10^4^	-	-
Longitudinal Bar	HRB400	400	*E_st_*_1_ = 2.0 × 10^5^	2.5	-
Stirrup	HPB300	300	*E_st_*_2_ = 2.1 × 10^5^	-	25–400

**Table 2 materials-17-03217-t002:** Parameters of each specimen.

Type	Specimen Number	Longitudinal Bar	Stirrup	Adopted Quantity
P	P-0	1Φ16	-	3
	P-100	1Φ16	φ6@100	3
	P-70	1Φ16	φ6@70	3
	P-50	1Φ16	φ6@50	3
R	R-0	1Φ16	-	3
	R-100	1Φ16	φ6@100	3
	R-70	1Φ16	φ6@70	3
	R-50	1Φ16	φ6@50	3

Note: “P” stands for plain round bar, and “R” stands for ribbed steel bar.

**Table 3 materials-17-03217-t003:** Concrete mixture rate.

Water–Cement Ratio	Cement/kg/m^3^	Water/kg/m^3^	Sand/kg/m^3^	Coarse Aggregate/kg/m^3^
0.53	375	200	750	1125

**Table 4 materials-17-03217-t004:** The compressive strength of the concrete after curing for 28 days.

Specimen 1/kN	Specimen 2/kN	Specimen 3/kN	Average Value/kN	Strength Value/[MPa]
542.9	611.7	595.4	583.3	55.42

**Table 5 materials-17-03217-t005:** The mechanical properties of the steel rebars.

Type	Nominal Diameter /mm	Yield Load /kN	Yield Strength /[MPa]	Elasticity Modulus /×10^5^ [MPa]
HPB300	16	68.28	357.23	2.12
	6	10.61	351.41	2.07
HRB400	16	100.91	571.06	2.26

**Table 6 materials-17-03217-t006:** Critical corrosion rate of the plain round bar at the cracking time with different spacings between the stirrups.

Spacing between Stirrups/mm	c/mm	d/mm	Stirrup Ratio	*f_c_*/[MPa]	*f_ct_*/[MPa]	*E_c_*/GPa	Actual Measured Corrosion Rate/%	Calculated Corrosion Rate/%	Deviation/%
								*n* = 2.5	
-	15	16	0	44.328	3.343	35.382	0.628	0.641	2.03
100	15	16	0.0957	44.328	3.343	35.382	0.815	0.732	11.37
70	15	16	0.137	44.328	3.343	35.382	0.884	0.774	14.17
50	15	16	0.191	44.328	3.343	35.382	0.806	0.832	3.14

**Table 7 materials-17-03217-t007:** Critical corrosion rate of the ribbed steel bar at the cracking time with different spacings between the stirrups.

Spacing between Stirrups/mm	c/mm	d/mm	Stirrup Ratio	*f_c_*/[MPa]	*f_ct_*/[MPa]	*E_c_*/GPa	Actual Measured Corrosion Rate/%	Calculated Corrosion Rate/%	Deviation/%
								n = 2.5	
-	15	16	0	44.328	3.343	35.382	0.665	0.641	3.74
100	15	16	0.0957	44.328	3.343	35.382	0.651	0.732	11.07
70	15	16	0.137	44.328	3.343	35.382	0.710	0.774	8.27
50	15	16	0.191	44.328	3.343	35.382	0.810	0.832	2.64

**Table 8 materials-17-03217-t008:** The corrosion depth of the steel bar when the concrete cover cracks.

Scholar	c/mm	d/mm	Stirrup Ratio	*f_c_*/[MPa]	*f_ct_*/[MPa]	*E_c_*/GPa	The *x_cr_* from the Experiment/μm	Calculated *x_cr_*/μm	
								n = 2.5	n = 3
Andrade [18]	20	16	-	-	3.55	36	14.4–17.9	25.5	16.7
	30	16	-	-	3.55	36	21.3	32.6	20.9
Liu [19]	27	16	-	31.5	3.3	27	31.4	31.9	20.6
	48	16	-	31.5	3.3	27	51.6	55.8	34.7
	70	16	-	31.5	3.3	27	73.8	93.8	56.6
Zhou [34]	20	12	50	-	2.01	30	26.3–32.9	30.8	19.5
	20	12	70	-	2.01	30	19.9	29.1	18.6
	20	12	100	-	2.01	30	13.1–16.0	27.1	17.5
	20	12	150	-	2.01	30	13.1–16.0	25.6	16.7

Note: During the calculation, both the *f_ct_* and *E_c_* unknown in the test are calculated using the regression formula: *E_c_* = 100/(2.2 + 34.7/*f_cu_*)(GPa) and *f_ct_* = 0.23*f_cu_*^2/3^, *f_cu_* = *f_c_*/0.8.

**Table 9 materials-17-03217-t009:** The number of steel bars.

Diameter/mm	4.24	6	8.49
Spacing/mm			
Without stirrups	A		
100	B-1	B-2	B-3
70	C-1	C-2	C-3
50	D-1	D-2	D-3

**Table 10 materials-17-03217-t010:** The ratio relationship between different spacings and different sectional areas of the stirrups.

Spacing /mm	Sectional Area/mm^2^	The Value of the Theoretical Model	The Value of ABAQUS	Deviation/%
100	18	1.24	1.12	9.41
	36	1.44	1.18	18.16
	72	1.98	1.35	32.11
70	18	1.35	1.19	12.01
	36	1.66	1.32	20.36
	72	2.20	1.51	31.42
50	18	1.44	1.24	14.06
	36	1.98	1.72	13.10
	72	3.13	2.82	10.05

## Data Availability

The data presented in this study are available on request from the corresponding author due to scientific research reason.

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
