# Peer review of "Corrosion-Induced Cracking Model of Concrete Considering a Transverse Constraint"

_materials, 2024, doi:10.3390/ma17133217_

Round 1
Reviewer 1 Report
Comments and Suggestions for Authors
The topic investigated holds significant interest within the research community, particularly given the recent surge in extreme events worldwide. The content is well-structured, and well written. As a result, I wholeheartedly recommend its acceptance for publication in Materials but after some modifications.
However, I do have a few minor comments and suggestions for the authors aimed at further enhancing the manuscript's quality. While the paper is commendable in its current form, these suggestions are intended to refine it further and ensure its excellence.
- The authors state that '50 years in China' is a fixed time from the national code. However, since the title and research are intended for a global audience, it would be beneficial to extend this statement to include other countries. I suggest mentioning that a 50-year period is also commonly used in other nations, to better reflect the worldwide applicability of the research.
- Authors extensively along the paper use “et al[18]”. I suppose the right format is “et al. [18]”
- Figure 1 seems to come from a different paper/source. Is it needed to write in the caption something like “adapted from” or “copyright given from” or similar ? As an alternative I can suggest to re-draw the figures.
- I suggest to improve quality of Figure 4.
- Author use MPa but I suggest, to be consistent with the International System, to use [MPa]
- Software ABAQUS requires a reference to be added in the bibliographic section. Please add the type of finite elements used. From figures they seem C3D8R (in this case please add a reference to explain the use of these elements such as DOI 10.1515/cls-2022-0027)
- Compared to the amount of analyses done “Conclusion” section is a bit poor. I suggest the authors to introduce more information.
Author Response
Reviewer 1
The topic investigated holds significant interest within the research community, particularly given the recent surge in extreme events worldwide. The content is well-structured, and well written. As a result, I wholeheartedly recommend its acceptance for publication in Materials but after some modifications.
However, I do have a few minor comments and suggestions for the authors aimed at further enhancing the manuscript's quality. While the paper is commendable in its current form, these suggestions are intended to refine it further and ensure its excellence.
Suggestion 1:
The authors state that “50 years in China” is a fixed time from the national code. However, since the title and research are intended for a global audience, it would be beneficial to extend this statement to include other countries. I suggest mentioning that a 50-year period is also commonly used in other nations, to better reflect the worldwide applicability of the research.
Answer: Accept and thanks for your rigorous suggestion. This sentence has been revised to “Therefore, in accordance with the Code for Design of Concrete Structures and other worldwide standards, reinforced concrete structures are supposed to show good service performance for at least 50 years in China and other countries worldwide”.
Suggestion 2:
Authors extensively along the paper use “et al[18]”. I suppose the right format is “et al. [18]”
Answer: The suggestion is accepted. We have checked and modified all the “et al” in the full text.
Suggestion 3:
Figure 1 seems to come from a different paper/source. Is it needed to write in the caption something like “adapted from” or “copyright given from” or similar? As an alternative I can suggest to re-draw the figures.
Answer: Thank you for your suggestion. Figure 1 to Figure 4 have all been redrawn and the image quality has been improved.
Suggestion 4:
I suggest to improve quality of Figure 4.
Answer: The suggestion is accepted. Figure 4 has been redrawn and the image quality has been improved.
Suggestion 5:
Author use MPa but I suggest, to be consistent with the International System, to use [MPa]
Answer: The suggestion is accepted. We have checked and modified all the “MPa” in the full text.
Suggestion 6:
Software ABAQUS requires a reference to be added in the bibliographic section. Please add the type of finite elements used. From figures they seem C3D8R (in this case please add a reference to explain the use of these elements such as DOI 10.1515/cls-2022-0027)
Answer: The suggestion is accepted. An explanation has been added in Section 4.1, “The type of finite elements in this model is C3D8R. C3D8R is an eight-node linear hex-ahedral element, which is the least time-consuming hexahedral element in ABAQUS, so it is the most widely used in the practical application of volume element.” Your recommended literature is very helpful, and references have been added.
Suggestion 7:
Compared to the amount of analyses done “Conclusion” section is a bit poor. I suggest the authors to introduce more information.
Answer: Accept. The “Conclusions” section has been modified. Please review in the revised manuscript.
Reviewer 2 Report
Comments and Suggestions for Authors
In this study a three-layer hollow cylinder model was developed to estimate the critical level of steel bar corrosion at which concrete cover begins to crack. This model also considers the effect of stirrups constraining the surrounding concrete, enabling it to predict the corrosion rate in members with stirrups at the time concrete cover cracks. The paper is engaging, but it needs some refinements before it can be published.
1. Chapter 1, Introduction requires a more detail analysis on the recent proposal in the field of concrete technology to deal with the challenges due to extensive corrosion (https://doi.org/10.1016/j.conbuildmat.2024.135491) as well as numerical technologies developed for mechanical and environmental characterization of (https://doi.org/10.3390/app11052000) concrete structures.
2. The available gaps in the literature providing the motivation of the study must clearly be stated in the end of the chapter 1.
3. Quality of the figures 1 to 4 and the associated explanations must be enhanced.
4. The elements demonstrated in figure 8 may be explained in the body of the paper.
5. A technical explanation of the information provided in the Tables 6 and 7 is required.
6. Assumption of the finite element model including the number of elements and mesh type can be discussed.
7. Limitation of the study and the possible future research must be discussed.
8. Conclusion of the paper must include the main achievements of the study, thus, it must be rewritten.

Author Response
Reviewer 2
In this study a three-layer hollow cylinder model was developed to estimate the critical level of steel bar corrosion at which concrete cover begins to crack. This model also considers the effect of stirrups constraining the surrounding concrete, enabling it to predict the corrosion rate in members with stirrups at the time concrete cover cracks. The paper is engaging, but it needs some refinements before it can be published.
Suggestion 1:
Chapter 1, Introduction requires a more detail analysis on the recent proposal in the field of concrete technology to deal with the challenges due to extensive corrosion (https://doi.org/10.1016/j.conbuildmat.2024.135491) as well as numerical technologies developed for mechanical and environmental characterization of (https://doi.org/10.3390/app11052000) concrete structures.
Answer: The suggestion is accepted. Thank you for recommending helpful literatures, they have been added as references, “However, the material components of concrete will react with CO2 and water under natural environment which may reduce the alkalinity of concrete[2].”, and “So the possibility of corrosion for steel bars in concrete is increasing with time[3]”.
Suggestion 2:
The available gaps in the literature providing the motivation of the study must clearly be stated in the end of the chapter 1.
Answer: The suggestion is accepted. At the end of Chapter 1, a summary of the defects in the existing researches has been added, and what work has been done in this paper to make up for these gaps has been pointed out. Please view it in the revised manuscript.
Suggestion 3:
Quality of the figures 1 to 4 and the associated explanations must be enhanced.
Answer: Thank you for your suggestion. Figure 1 to Figure 4 have all been redrawn and the image quality has been improved. Associated explanations of Figure 1~Figure 4 has been added to the text.
Suggestion 4:
The elements demonstrated in figure 8 may be explained in the body of the paper.
Answer: The suggestion is accepted. An explanation has been added in Section 3.2, “The galvanic accelerated corrosion method is shown in Figure 8. Cover the concrete block with a sponge and secure the sponge with a stainless-steel mesh. A layer of insulating plastic cloth is covered outside the stainless-steel mesh to ensure that the concrete remains wet during the test. The positive pole of the DC power supply is connected to the exposed steel bar, and the negative pole is connected to the stainless-steel mesh. The reason for this connection is that electrons flow from the negative to the positive side of DC power. As the anode, the steel bar loses electrons, so the electrons enter the stain-less-steel mesh and flow out from the exposed section of the steel.”.
Suggestion 5:
A technical explanation of the information provided in the Tables 6 and 7 is required.
Answer: The suggestion is accepted. An explanation has been added for Tables 6 and 7 in Section 3.3:
“Since the diameter, stirrup ratio, mechanical properties and other indicators of ribbed steel bars and plain round bars are exactly the same, their calculated critical corrosion rates are the same, all between 0.641%~0.832%. When the concrete cover cracks, the measured critical corrosion rate of the plain round bar is between 0.628%~0.884%. For ribbed steel bars, the measured critical corrosion rate is between 0.651%~0.810%. It is found that the actual measured corrosion rate is discrete. The main reason is that the number of specimens is too small. However, since the deviation is basically within 20%, the calculations are still convincing. In this test, the longitudinal bar is always free from load during the process of corrosion. By comparing the actual measured data in two tables, it can be seen that when the longitudinal bar is not loaded, the critical corrosion rate of the concrete with plain round bar is basically the same as that of the concrete with ribbed steel bar.
By comparing the data in Table.6 and 7, it can be found that when there is no stir-rup in concrete, the model can almost fully explain the actual state. The same situation happens when the spacing between the stirrups is about 50mm or less. This shows that the assumptions of the model and its theoretical derivation are correct. When the spacing between the stirrups was between about 50mm and 100mm, the model deviated slightly from the actual state. This is because the model assumes that the stirrup is continuous longitudinally. If the spacing between stirrups is too large, the assumption will not hold. This critical value of the spacing is discussed in detail in the finite element analysis section.”
Please see the detailed explanation in Section 3.3 of the manuscript.
Suggestion 6:
Assumption of the finite element model including the number of elements and mesh type can be discussed.
Answer: The suggestion is accepted. An explanation has been added in Section 4.1, “The type of finite elements in this model is C3D8R. C3D8R is an eight-node linear hex-ahedral element, which is the least time-consuming hexahedral element in ABAQUS, so it is the most widely used in the practical application of volume element. The model contains 26500 elements, in which the element density is increased at the steel-concrete interface to improve the calculation accuracy.”
Suggestion 7:
Limitation of the study and the possible future research must be discussed.
Answer: The suggestion is accepted. We have discussed the limitations and possible future research directions of this model at the end of Section 4.2. Please review in the revised manuscript.
Suggestion 8:
Conclusion of the paper must include the main achievements of the study, thus, it must be rewritten.
Answer: The suggestion is accepted. The “Conclusions” section has been modified. Please review in the revised manuscript.
Reviewer 3 Report
Comments and Suggestions for Authors
Interesting research and modeling of corrosion cracks in reinforced concrete. Well described experiment. ABAQUS model evaluted by comparison with laboratory results. Paper is interesting for the readers and could be published after mainor improvements:
1. Table 3 - very high content of water in concrete - why there is no water reducing admixture in the recipe? This mix composition is not usable in modern concrete work on site. It is neccessary to justify the reason of SP elimination
2. tab 3 - what do you mean by "stone"? This term is not appropriate - do you mean gravel or grit? you may also use a term "coarse aggregate"
3. tab.4 What is "corrected value"? please explain in the text or under the table
4. tabl 6 an d 7 - what are the differences between this tables? Titles are identical - it is not understandable what was the variable (steel type?)
5. Some interesting sources for the Authors could be find in publications. - for example
A cracking model for reinforced concrete cover, taking account of the accumulation of corrosion products in the ITZ layer, and including computational and experimental verification, Materials, MDPI, vol. 13, nr 23, 2020, 5375, 1-17, DOI:10.3390/ma13235375
Author Response
Reviewer 3
Interesting research and modeling of corrosion cracks in reinforced concrete. Well described experiment. ABAQUS model evaluted by comparison with laboratory results. Paper is interesting for the readers and could be published after mainor improvements:
Suggestion 1:
Table 3 - very high content of water in concrete - why there is no water reducing admixture in the recipe? This mix composition is not usable in modern concrete work on site. It is neccessary to justify the reason of SP elimination
Answer: Thank you for your advice. Since our research group has carried out many researches on concrete with water-binder ratio of 0.53 and 0.35, this study follows the previous water-binder ratio in order to form a reference and comparison with previous researches. In addition, this study did not add water reducing admixture, in order to reduce the types of material, as far as possible to reduce the influence of additives on the mechanical properties of concrete. We have been carrying out research on the durability of concrete added with water reducing admixtures, which will appear in future submissions.
Suggestion 2:
tab 3 - what do you mean by "stone"? This term is not appropriate - do you mean gravel or grit? you may also use a term "coarse aggregate"
Answer: The suggestion is accepted. Thank you for your suggestion. The word “stone” has been revised to “coarse aggregate”.
Suggestion 3:
tab.4 What is "corrected value"? please explain in the text or under the table
Answer: The suggestion is accepted. The phrase “corrected value” means the compressive strength of concrete after correction due to the size effect, following the Chinese standard GB/T50081-2002. The “corrected value”is deleted as it can generate confusion to the reading.
Suggestion 4:
tabl 6 and 7 - what are the differences between this tables? Titles are identical - it is not understandable what was the variable (steel type?)
Answer: These two tables show the critical corrosion rates of plain round bar and ribbed steel bars, respectively. Table 6 shows the critical corrosion rate of plain round bar at cracking time of different spacing between stirrups, while Table 7 shows the critical corrosion rate of ribbed steel bar.
Suggestion 5:
Some interesting sources for the Authors could be find in publications. - for example
Krykowski Tomasz, Jaśniok Tomasz, Recha Faustyn, Karolak M.: A cracking model for reinforced concrete cover, taking account of the accumulation of corrosion products in the ITZ layer, and including computational and experimental verification, Materials, MDPI, vol. 13, nr 23, 2020, 5375, 1-17, DOI:10.3390/ma13235375
Answer: Thank you for your recommendation. This paper is very helpful to us. We have added a reference of this paper. Please find this paper in the list of references.
Round 2
Reviewer 2 Report
Comments and Suggestions for Authors
The authors have completely applied all the proposed comments, thus the paper can proceed with the publication.